# Lung ultrasound score to monitor COVID-19 pneumonia progression in patients with ARDS

**Auguste Dargent**[1]*, **Emeric Chatelain**[1], **Louis Kreitmann**[1,2], **Jean-Pierre Quenot**[3,4,5,6], **Martin Cour**[1,2], **Laurent Argaud**[1,2], **the COVID-LUS study group**[¶]

1 Service de Médecine Intensive-Réanimation Médicale, Hospices Civils de Lyon, Hôpital Edouard Herriot, Lyon, France, 2 Faculté de médecine Lyon-Est, Université de Lyon, Université Claude Bernard Lyon 1, Lyon, France, 3 Médecine Intensive Réanimation, CHU Dijon, Dijon, France, 4 Université Bourgogne Franche-Comté, LNC, Dijon, France, 5 INSERM, LNC UMR1231, Dijon, France, 6 FCS Bourgogne-Franche Comté, LipSTIC LabEx, Dijon, France

¶ Membership of the COVID-LUS study group is listed in the Acknowledgments.
* auguste.dargent@chu-lyon.fr

**Data Availability Statement:** All relevant data are within the manuscript and its Supporting Information.

## Abstract

COVID-19 pneumonia typically begins with subpleural ground glass opacities with progressive extension on computerized tomography studies. Lung ultrasound is well suited to this interstitial, subpleural involvement, and it is now broadly used in intensive care units (ICUs). The extension and severity of lung infiltrates can be described numerically with a reproducible and validated lung ultrasound score (LUSS). We hypothesized that LUSS might be useful as a tool to non-invasively monitor the evolution of COVID-19 pneumonia at the bedside. LUSS monitoring was rapidly implemented in the management of our COVID-19 patients with RT-PCR-documented COVID-19. The LUSS was evaluated repeatedly at the bedside. We present a graphic description of the course of LUSS during COVID-19 in 10 consecutive patients admitted in our intensive care unit with moderate to severe ARDS between March 15 and 30th. LUSS appeared to be closely related to the disease progression. In successfully extubated patients, LUSS decreased and was lower than at the time of intubation. LUSS increased inexorably in a patient who died from refractory hypoxemia. LUSS helped with the diagnosis of ventilator-associated pneumonia (VAP), showing an increased score and the presence of new lung consolidations in all 5 patients with VAPs. There was also a good agreement between CT-scans and LUSS as for the presence of lung consolidations. In conclusion, our early experience suggests that LUSS monitoring accurately reflect disease progression and indicates potential usefulness for the management of COVID-19 patients with ARDS. It might help with early VAP diagnosis, mechanical ventilation weaning management, and potentially reduce the need for X-ray and CT exams. LUSS evaluation is easy to use and readily available in ICUs throughout the world, and might be a safe, cheap and simple tool to optimize critically ill COVID-19 patients care during the pandemic.

## Introduction

According to computerized tomography (CT) studies, COVID-19 pneumonia typically begins with subpleural ground glass opacities with progressive extension [1]. Lung ultrasound is well

**Funding:** The author(s) received no specific funding for this work.

**Competing interests:** The authors have declared that no competing interests exist.

suited to this interstitial, subpleural involvement, and it is now broadly used in intensive care units (ICUs) [2]. Typical lung ultrasound of COVID-19 pneumonia reveals a pattern of B-lines (i.e. comet-tail artifacts perpendicular to the pleural surface), which may spread to the whole pleura in severe patients [3]. The extension and severity of lung infiltrates can be described numerically with a reproducible and validated lung ultrasound score (LUSS) [4]. However, this score has not been widely and routinely used for ARDS monitoring so far. The typical, "lobar" ARDS pattern is driven by posterior consolidations weakening the diagnostic accuracy of the score [5], whereas COVID-19 patients exhibit a different (spatial and temporal) course of disease, with a diffuse and rapidly evolving pattern [1]. We hypothesized that LUSS might be useful as a tool to non-invasively monitor the evolution of COVID-19 pneumonia at the bedside. We present here a brief report of our early experience of LUSS use.

## Patients and methods

LUSS monitoring was rapidly implemented in the management of our COVID-19 patients with RT-PCR-documented COVID-19 pneumonia, especially when requiring mechanical ventilation. Every practitioner in the unit was trained and LUSS was integrated to the daily routine examination. Junior physicians were trained at the bedside, and were accompanied by a trained investigator until a good interobserver agreement was reached (which required about 4–5 LUSS). The LUSS was evaluated at the bedside, as previously described [4]. Briefly, 0–3 points were allocated for each of the 12 pre-determined anatomical regions according to ultrasound pattern: normal = 0, well-defined B-lines = 1, coalescent B-lines = 2, consolidation = 3 (total score ranging from 0 to 36). Lung consolidations (scoring 3) were noted only when thickness (perpendicular from pleura) was greater than 15 mm. Sub-pleural thickening and sub-pleural consolidations (thickness 15 mm or thinner) were graded a score of 2. Continuous variables are expressed as median (interquartile range). The result of each score with the detail of each zone was recorded in a table and kept in the patient medical record, allowing the monitoring from day-to-day. The LUSS was evaluated daily, when possible (depending on workload and available medical staff).

This study was approved by our institutional ethics committee (*Comité d'éthique du CHU de Lyon*) and was conducted in accordance with the 1964 Helsinki declaration and its later amendments. Patients and/or relatives gave informed consent for the collection of clinical data.

## Results

We report in Fig 1 the course of LUSS during COVID-19 in 10 consecutive patients admitted in our ICU with moderate to severe ARDS between March 15 and 30[th]. Clinical characteristics of these patients were as follows: age 56 (46–63) years, 80% men, body-mass index 32 (30–33), simplified acute physiology score II 36 (34–38). A total of 109 LUSS were assessed for these patients (with 10.9 distinct LUSS evaluations per patient on average). The LUSS at ICU admission was 22 (15–26), maximum and lowest LUSS were 27 (26–29) and 14 (12–15), respectively. During the study period, 4 patients were successfully weaned from mechanical ventilation (MV). In all of them, LUSS on the day of extubation was all lower than admission LUSS, with a LUSS within 24h of extubation ranging from 13 to 17. Patient #6 was also extubated but was not considered as weaned from mechanical ventilation as he presented an early post-extubation acute respiratory failure requiring high-intensity non-invasive ventilation. His LUSS on the day of extubation was of 27. LUSS increased inexorably in the patient who died from refractory hypoxemia. In the five patients presenting a ventilator-associated pneumonia (VAP), lung ultrasound monitoring showed an increased LUSS and especially the presence of

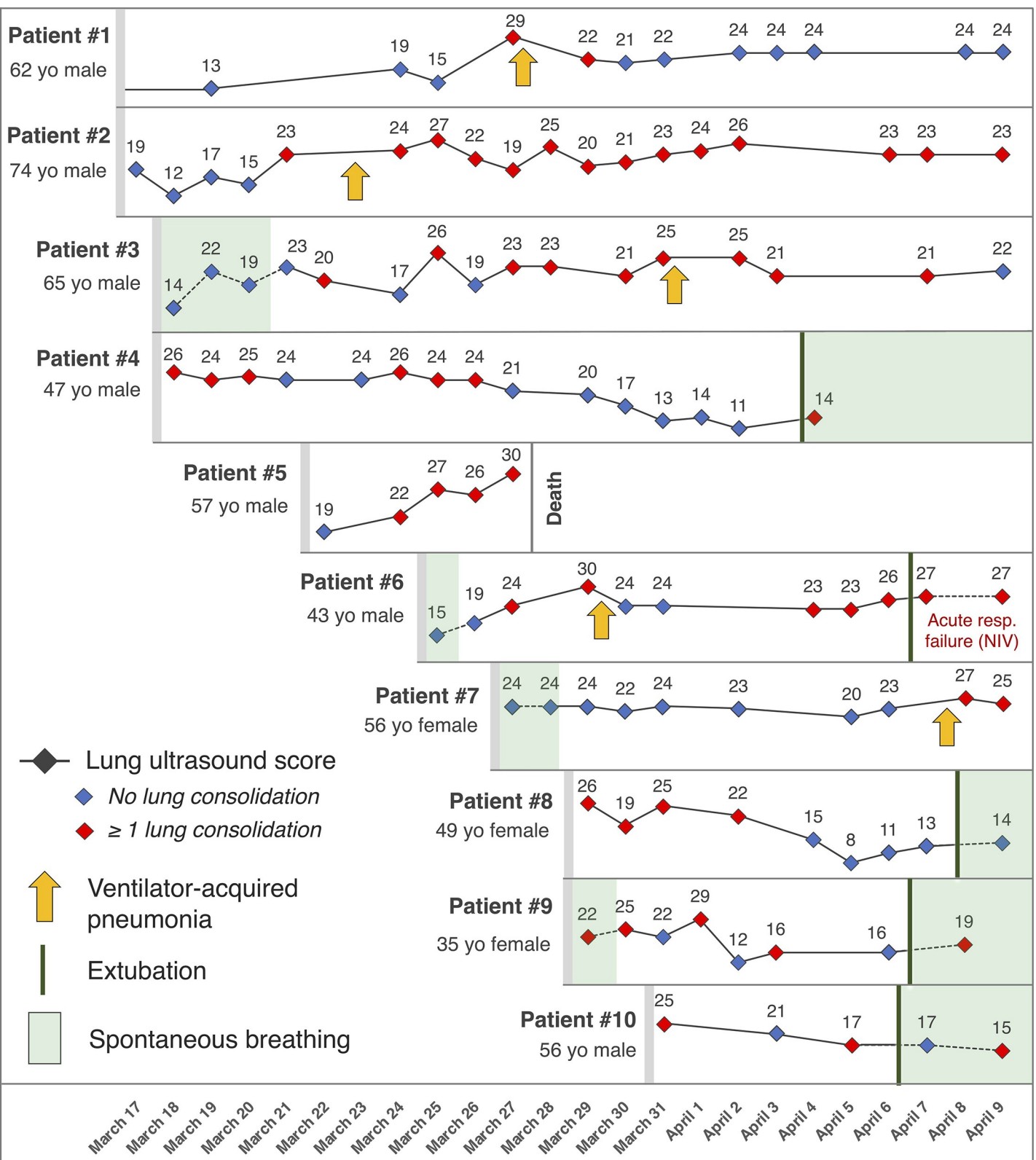

**Fig 1. Clinical course and lung ultrasound score (LUSS) evolution in 10 COVID-19 patients.** For each timepoint the numerical LUSS is indicated above the corresponding symbol (blue diamond if ultrasound showed only interstitial infiltrates and red if it also showed at least one lung consolidation). Arrows indicate occurrence of a clinically-diagnosed ventilator-associated pneumonia. Mechanical ventilation-free periods are highlighted in green. Patient #5 died of refractory

hypoxemia. Patient #6 was extubated on April 7[th] but was placed under non-invasive ventilation due to post-extubation acute respiratory failure. Patients #4 and #9 were discharged alive from intensive care during the study period. NIV: non-invasive ventilation; yo: years-old.

new lung consolidation. Indeed, such new consolidations (zone-score of 3) were detected in the patients with VAPs in one or two zones (often lateral-inferior or posterior-inferior) which were free of consolidation in the previous ultrasound examinations (zone-score of 0, 1 or 2). VAPs were diagnosed with the usual criteria, and the day of VAP occurrence was defined as the day when positive tracheal aspirate or bronchoalveolar fluid was sampled (all VAP episodes were culture-positive). CT-scans were performed in 7 patients and all showed diffuse ground glass opacities. Four CT-scans also found lung consolidations, which were all detected by ultrasonography.

## Discussion

Our early experience with LUSS monitoring indicates potential usefulness for the management of COVID-19 patients with ARDS. First, our results suggest that LUSS could contribute to the early diagnosis of VAPs. Lung ultrasound was previously described as a useful tool for VAP diagnosis [6], and it might be well suited in the setting of COVID pandemic, where early and/or non-superinfected cases are typically free of lung consolidations [1]. One major limitation of LUSS in this setting is that a zone-score of 3 does not allow to compare the size of an existing consolidation from day-to-day, preventing a more subtle interpretation of this capital information. Second, we observed that all patients successfully weaned from MV had a decreased LUSS. Of note, LUSS was validated as a mechanical ventilation-weaning prediction tool [4], even though the cut-off value remains to be determined for COVID-19 patients. In all cases, LUSS should not be interpreted in isolation but rather integrated to the set of usual parameters when considering a therapeutic decision.

Our early experience so far encouraged us to continue and use LUSS in our routine practice for COVID-19 patients, as it seems to accurately reflect disease progression. LUSS evaluation is easy to use and readily available in ICUs throughout the world, and might be a safe, cheap and simple tool to optimize critically ill COVID-19 patients care during the pandemic. Further studies are now needed to determine if LUSS-based interventions can effectively improve the management of COVID-19 patients with ARDS, e.g. by allowing faster MV weaning, earlier detection and treatment of VAPs, or optimizing care efficiency by reducing the need for X-ray and CT exams.

## Supporting information

**S1 Data.**
(XLSX)

## Acknowledgments

### Members of the COVID-LUS study group

Meghann Antoine[1], Thomas Baudry[1], *, Pierre-Jean Bertrand[1], Joanna Bougnaud[1], François Charbonnieras[1], Matthieu Damien[1], Camille Delsol[1], Pierre Gimet[1], Alan Moutou[1], Emilie Nourry[1], Alexandre Pinede[1], Zoé Lougnon[1], Hugo Roccia[1], Marine Romain[1], Marie Simon[1], Karine Srage[1], Matthieu Venet[1]

* Study group lead author

Email: thomas.baudry@chu-lyon.fr

[1] Service de Médecine Intensive-Réanimation Médicale, Hospices Civils de Lyon, Hôpital Edouard Herriot, Lyon, France

## Author Contributions

**Conceptualization:** Auguste Dargent.

**Data curation:** Auguste Dargent.

**Investigation:** Auguste Dargent, Emeric Chatelain, Louis Kreitmann.

**Supervision:** Auguste Dargent, Jean-Pierre Quenot.

**Validation:** Martin Cour.

**Writing – original draft:** Auguste Dargent, Emeric Chatelain, Martin Cour.

**Writing – review & editing:** Auguste Dargent, Emeric Chatelain, Louis Kreitmann, Jean-Pierre Quenot, Martin Cour, Laurent Argaud.

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
