## [Decision Letter · Decision Letter 0]

10 Jun 2020

PONE-D-20-14729

Lung ultrasound score to monitor COVID-19 pneumonia progression in patients with ARDS

PLOS ONE

Dear Dr. Auguste Dargent,

Thank you for submitting your manuscript to PLOS ONE. After careful consideration, we feel that it has merit but does not fully meet PLOS ONE’s publication criteria as it currently stands. Therefore, we invite you to submit a revised version of the manuscript that addresses the points raised during the review process.

I have received the comments of the reviewers on your manuscript. The specific comments of the reviewers are included below. Please provide point by point response in your revised manuscript.

We look forward to receiving your revised manuscript.

Kind regards,

Muhammad Adrish

Academic Editor

PLOS ONE

Journal Requirements:

2. One of the noted authors is a group, COVID-LUS study group.

In addition to naming the author group, please list the individual authors and affiliations within this group in the acknowledgments section of your manuscript.

Please also indicate clearly a lead author for this group along with a contact email address.

Reviewers' comments:

Reviewer's Responses to Questions

**Comments to the Author**

1. Is the manuscript technically sound, and do the data support the conclusions?

Reviewer #1: Yes

Reviewer #2: Partly

Reviewer #3: No

2. Has the statistical analysis been performed appropriately and rigorously? 

Reviewer #1: N/A

Reviewer #2: N/A

Reviewer #3: No

3. Have the authors made all data underlying the findings in their manuscript fully available?

Reviewer #1: Yes

Reviewer #2: Yes

Reviewer #3: Yes

4. Is the manuscript presented in an intelligible fashion and written in standard English?

Reviewer #1: Yes

Reviewer #2: Yes

Reviewer #3: Yes

5. Review Comments to the Author

Reviewer #1: The article describes monitoring of pneumonia progression in patients with COVID-19 infection using lung ultrasound.

It is well written and interesting.

Lung ultrasound is an established approach for monitoring pulmonary diseases in the ICU. It makes a lot of sense to use this method in this population of COVID-19 infected patients from an infection control perspective. It is a bedside test and can be performed rather quickly. If done by experienced providers, it can save trips to the CT which always comes at a risk of infecting other patients or staff if the patient traveling for the test has a contagious disease.

The provided figure gives a good overview of score development during the hospitalization and puts it in clinical perspective.

My only critique is that it would be very helpful for the reader to have a panel of pictures from a patient prior and post development of VAP to illustrate the described changes as the authors promote this method for earlier diagnosis of VAP in COVID-19 patients. This is particularly important as there is currently debate about the difference of ARDS in patient with COVID-19 amongst themself and compared to the traditional population diagnosed with ARDS.

Reviewer #2: I do want to congratulate the authors on their work. This is an interesting paper looking at the LUSS score in the management of the COVID 19 patients.

Using of Lung US in well documented in the critical care practice. Use of LUSS score has been used in management of ARDS.

Although study is interesting there are many limitations to study. First the patient number is small to draw any definite conclusions

Second there no clear mythological pattern describe by author , as daily or twice a day LUSS score assessment.

Third: In patient outcomes, it would be ideal to know what was the LUSS score prior to extubation if that is available, as there are studies that have showed correlation between the two. Ref : Caltabeloti F, Monsel A, Arbelot C, Brisson H, Lu Q, Gu WJ, et al. Early fluid loading in acute respiratory distress syndrome with septic shock deteriorates lung aeration without impairing arterial oxygenation: a lung ultrasound observational study. Crit Care 2014;18:R91.

Fourth : The author have showed LUSS score >1 region with score of 3 and then marked that point as VAP in 5 of their patients , it would be beneficial to know if these ultrasound finding also correlated with clinically diagnosis of VAP.

Fifth: It has been established that score of 3 can have limitations as that if the area consolation is not in taken into account, it can overestimate the lack of aeration and give a high score, I would like if authors could justify if they took this into consideration. Ref: Chiumello D, Mongodi S, Algieri I, Vergani GL, Orlando A, Via G, et al. Assessment of lung aeration and recruitment by CT scan and ultrasound in acute respiratory distress syndrome patients. Crit Care Med 2018;46:1761–1768.

Overall interesting concept and may be clinically useful tool to guide therapy in treatment of this new disease.

Reviewer #3: General

In this manuscript, Dargent et al describe serial assessments of the Lung Ultrasound Score (LUSS) on 10 consecutive patients admitted to their intensive care unite with moderate to severe ARDS. Multiple scores were available for each patient (average of 10.9 scans per patient on average) and their course of LUSSs were tracked throughout the study. They found that in successfully extubated patients, the LUSS decreased and was lower than at the time of intubation. In a patient who died from refractory hypoxemia the LUSS increased. They suggest that the LUSS accurately reflects disease progression and may help in early VAP diagnosis, mechanical ventilation weaning management, and potentially reduce the need for other imaging.

Overall I was intrigued by the apparent association in lower scores in those who were successfully weaned from mechanical ventilation, but I don’t have strong confidence in the validity of this claim given the unknown result of patient #6 (see below). Whether patient #6 needed non-invasive ventilation or was re-intubated it was associated with a high LUSS indicating that extubation may have been premature.

Major

A PubMed entry of "COVID Lung US" yielded 243 results – as such, this study has missed the opportunity to be presented as “pioneering” work, and as such a description of 10 patients without the ability to clearly demonstrate added value beyond already available clinical data, and without any statistical testing at all fundamentally limits my enthusiasm.

The short term and variable length of followup is problematic. Figure 1 shows this problem graphically and makes it hard to draw any meaningful conclusions about the observations described.

It’s not clear how the results of the LUSS affected care, particularly with the lack of a control group. It’s impossible to draw conclusions of benefit from this manuscript.

Patient #6 was extubated on April 7th with a LUSS of 27, but placed on non-invasive ventilation. It is not mentioned if this patient was re-intubated or not, but if the patient was not I wonder if the group is wrong to which the patient is assigned. If not re-intubated then the patient was breathing spontaneously on non-invasive ventilation and by some would be considered to have been successfully extubated. Was the patient not included in this group because of re-intubation or because the LUSS would not allow a claim of lower LUSS in those successfully extubated because of the LUSS at that time being 27 or because the authors felt that the need for non-invasive ventilation does not meet the criteria of a successful wean from mechanical ventilation?

A claim is made that the LUSS decreased for those patients who were successfully extubated, but I see no statistical analysis of the LUSS to see if there was actually a statistical difference.

Minor

I cannot find the number of sonographers reported who performed the exams. Given the operator dependent nature of ultrasonography there is a potential for issues with inter-observer reliability. This may have led differences in LUSS not due to a change in disease severity, but purely because another sonographer scanned the patient that day.

The method by which the diagnosis of VAP was reached was not outlined in the study or whether the sonographer was blinded to this or if they were the ones making the diagnosis.

6. PLOS authors have the option to publish the peer review history of their article (what does this mean?). If published, this will include your full peer review and any attached files.

Reviewer #1: Yes: Andreas Schmid

Reviewer #2: No

Reviewer #3: No

---

## [Author Response · Author response to Decision Letter 0]

16 Jun 2020

Reviewer #1: The article describes monitoring of pneumonia progression in patients with COVID-19 infection using lung ultrasound.

It is well written and interesting.

Lung ultrasound is an established approach for monitoring pulmonary diseases in the ICU. It makes a lot of sense to use this method in this population of COVID-19 infected patients from an infection control perspective. It is a bedside test and can be performed rather quickly. If done by experienced providers, it can save trips to the CT which always comes at a risk of infecting other patients or staff if the patient traveling for the test has a contagious disease.

The provided figure gives a good overview of score development during the hospitalization and puts it in clinical perspective.

The authors thank the reviewer for these comments.

My only critique is that it would be very helpful for the reader to have a panel of pictures from a patient prior and post development of VAP to illustrate the described changes as the authors promote this method for earlier diagnosis of VAP in COVID-19 patients. This is particularly important as there is currently debate about the difference of ARDS in patient with COVID-19 amongst themself and compared to the traditional population diagnosed with ARDS.

We thank the reviewer for raising this point. Unfortunately, we did not save specific images for each patient, which would have been a very good illustration of the changes observed in a given patient along time. In all cases of ventilator-acquired pneumonia detected by ultrasonographic monitoring, it was evidenced by the apparition in one or two zones (often lateral or posterior) of a lung consolidation in a zone previously free of consolidation, within a few days’ interval. We are not able to show the actual images taken in the same patient, but we added a more precise description of the changes observed when a VAP was diagnosed. We also added details in the methods section as to how the score was recorded zone by zone. 

Reviewer #2: I do want to congratulate the authors on their work. This is an interesting paper looking at the LUSS score in the management of the COVID 19 patients.

Using of Lung US in well documented in the critical care practice. Use of LUSS score has been used in management of ARDS.

Although study is interesting there are many limitations to study. First the patient number is small to draw any definite conclusions

The authors absolutely agree with the reviewer #2. Our point is to give an overview of the use of this simple tool in the clinical practice during the pandemic. Even with the little number of patients presented here and managed later in our unit, we felt that ultrasound monitoring brought useful insight of the clinical evolution of these complex patients. That is why we wanted to share our experience with an individual, visual overview allowed by the numeric LUS score. 

Second there no clear mythological pattern describe by author , as daily or twice a day LUSS score assessment.

The LUS monitoring was implemented as a “real life” monitoring tool. The medical staff available and workload allowed daily monitoring for some periods but this was more difficult at other times (e.g. weekends). We agree that systematic screening with a fixed interval would have been ideal, although difficult to apply in this stressful period. A statement was added to the methods section.

Third: In patient outcomes, it would be ideal to know what was the LUSS score prior to extubation if that is available, as there are studies that have showed correlation between the two. Ref : Caltabeloti F, Monsel A, Arbelot C, Brisson H, Lu Q, Gu WJ, et al. Early fluid loading in acute respiratory distress syndrome with septic shock deteriorates lung aeration without impairing arterial oxygenation: a lung ultrasound observational study. Crit Care 2014;18:R91.

We fully agree with this comment. Indeed mechanical ventilation weaning is one major point where LUSS monitoring could be useful. We added the range of LUS scores of the extubated patients. The individual scores can be visualized on the figure (spontaneous breathing after successful extubation materialized by the green background color).

Fourth : The author have showed LUSS score >1 region with score of 3 and then marked that point as VAP in 5 of their patients , it would be beneficial to know if these ultrasound finding also correlated with clinically diagnosis of VAP.

Thank you for raising this point. The VAPs indicated by arrows in the figure were clinically diagnosed and treated. The arrow is placed on the day when positive tracheal aspirate or bronchoalveolar fluid was sampled. We made it clear by adding a statement in the legend of the figure and in the results section.

Fifth: It has been established that score of 3 can have limitations as that if the area consolation is not in taken into account, it can overestimate the lack of aeration and give a high score, I would like if authors could justify if they took this into consideration. Ref: Chiumello D, Mongodi S, Algieri I, Vergani GL, Orlando A, Via G, et al. Assessment of lung aeration and recruitment by CT scan and ultrasound in acute respiratory distress syndrome patients. Crit Care Med 2018;46:1761–1768.

We thank the reviewer for this very interesting comment. In the study cited by the reviewer, LUS 3 was not able to discriminate regions in which lung aeration was improved by PEEP, because the score 3 is given regardless of the size of the lung consolidation. This is indeed a major drawback of the LUS score, which was not taken into account in our study. For example, if a different physician is seeing the patient each day, he/she will not know if the consolidation size observed is decreased, increased or stable by just seeing “LUS 3” on the chart. We added a statement in the discussion section. 

Overall interesting concept and may be clinically useful tool to guide therapy in treatment of this new disease.

Reviewer #3: General

In this manuscript, Dargent et al describe serial assessments of the Lung Ultrasound Score (LUSS) on 10 consecutive patients admitted to their intensive care unite with moderate to severe ARDS. Multiple scores were available for each patient (average of 10.9 scans per patient on average) and their course of LUSSs were tracked throughout the study. They found that in successfully extubated patients, the LUSS decreased and was lower than at the time of intubation. In a patient who died from refractory hypoxemia the LUSS increased. They suggest that the LUSS accurately reflects disease progression and may help in early VAP diagnosis, mechanical ventilation weaning management, and potentially reduce the need for other imaging.

Overall I was intrigued by the apparent association in lower scores in those who were successfully weaned from mechanical ventilation, but I don’t have strong confidence in the validity of this claim given the unknown result of patient #6 (see below). Whether patient #6 needed non-invasive ventilation or was re-intubated it was associated with a high LUSS indicating that extubation may have been premature.

We thank the reviewer for raising this important point. The case of this particular patient is of interest and our description might be incomplete. Patient #6 was indeed extubated with a LUSS of 26 and presented an acute respiratory failure within 12 hours after extubation, requiring “rescue” non-invasive ventilation and high flow nasal cannula. That is why he is not included in “patients successfully weaned from mechanical ventilation”. This is explained in the figure legend. We added a statement in the result section. Furthermore, we modified the figure’s legend to make the outcome of this patient more clear.

Major

A PubMed entry of "COVID Lung US" yielded 243 results – as such, this study has missed the opportunity to be presented as “pioneering” work, and as such a description of 10 patients without the ability to clearly demonstrate added value beyond already available clinical data, and without any statistical testing at all fundamentally limits my enthusiasm.

As stated above, we wanted to give a rapid overview of the use of this simple tool in the clinical practice during the pandemic with this case series. Even with the little number of patients presented here and managed later in our unit, we felt that ultrasound monitoring brought useful insight of the clinical evolution of these complex patients. That is why we wanted to share our experience with an individual, visual overview allowed by the numeric LUS score. We agree that due to its small size and observational nature, this study does not allow to draw hard conclusions but rather an indication of its potential as a bedside tool. 

The short term and variable length of followup is problematic. Figure 1 shows this problem graphically and makes it hard to draw any meaningful conclusions about the observations described.

The authors thank the reviewer for this comment. We agree that the follow-up length were variable, although patients were admitted in a short period of time. We estimated that the clinical courses of our patients were indeed quite different but illustrated well the different complications and turning points where lung ultrasound can bring meaningful insight: clinical deterioration, weaning from mechanical ventilation, nosocomial pneumonia. 

We could now extend the follow-up duration for the patients to implement Figure 1, if necessary. However, we fear that it might overcrowd the Figure and make it less legible, without bringing more useful information. 

It’s not clear how the results of the LUSS affected care, particularly with the lack of a control group. It’s impossible to draw conclusions of benefit from this manuscript.

This is absolutely true. We cannot prove that patients benefited from LUSS monitoring. However, we felt that LUSS monitoring definitely improved clinical appraisal of the evolution of COVID-19 lung involvement. COVID-19 ARDS is a complex condition often challenging physicians with complex decisions to make, always based on multiple parameters like PaO2/FiO2, vitals, biomarkers. Our experience modestly suggests that LUSS can be one more parameter to take into account when diagnosing a ventilator acquired pneumonia or deciding whether to extubate a patient.

Patient #6 was extubated on April 7th with a LUSS of 27, but placed on non-invasive ventilation. It is not mentioned if this patient was re-intubated or not, but if the patient was not I wonder if the group is wrong to which the patient is assigned. If not re-intubated then the patient was breathing spontaneously on non-invasive ventilation and by some would be considered to have been successfully extubated. Was the patient not included in this group because of re-intubation or because the LUSS would not allow a claim of lower LUSS in those successfully extubated because of the LUSS at that time being 27 or because the authors felt that the need for non-invasive ventilation does not meet the criteria of a successful wean from mechanical ventilation?

This case of Patient #6 was already raised by the reviewer and is indeed interesting. This patient presented a post-extubation acute respiratory failure and required non-invasive ventilation and high flow nasal cannula. However, after a few days his condition improved after a corticosteroid therapy was started and he was not re-intubated. When NIV was finally weaned his LUSS was of 23 (not shown).

This is why we used the term “successfully weaned from mechanical ventilation” in the other patients, rather than “successful extubation”. The case of this patient was different because the extubation was not uneventful, and the fact that he had a high LUSS might be an explanation. 

A claim is made that the LUSS decreased for those patients who were successfully extubated, but I see no statistical analysis of the LUSS to see if there was actually a statistical difference.

We thank the reviewer for this comment. Given the small size of our patients’ sample, we did not perform statistical tests. By “decreased”, we meant that each patient’s LUSS was lower on the day of extubation as compared to the admission LUSS. We made that point clear in the result section.

Minor

I cannot find the number of sonographers reported who performed the exams. Given the operator dependent nature of ultrasonography there is a potential for issues with inter-observer reliability. This may have led differences in LUSS not due to a change in disease severity, but purely because another sonographer scanned the patient that day.

Thank you for raising this important point. Like we said, the LUSS was implemented in our ICU like a “real life”, shared tool. As such, every practitioner in the unit was trained and LUSS was integrated to the daily routine examination (overall by 20 junior and senior physicians). Junior physicians (including residents) were trained at the bedside, and were accompanied by a trained investigator until a good interobserver agreement was reached (about 4-5 LUSS). The reviewer is right about the inter-observer reliability issues that can arise from these multiple sonographers. However, we think it is also a strength of this experience, showing that a decent inter-operator congruence was obtained from such “real life” setting. This strengthens the external validity of this concept. Indeed, intensive care relies, by nature, on the intervention of multiple physicians. 

The method by which the diagnosis of VAP was reached was not outlined in the study or whether the sonographer was blinded to this or if they were the ones making the diagnosis.

This point is indeed important and was insufficiently adressed. The VAPs indicated by arrows in the figure were clinically diagnosed and treated. The arrow is placed on the day when positive tracheal aspirate or bronchoalveolar fluid was sampled. The sonographer was not really blinded to the diagnosis of VAP, but it was made independently from the LUSS examination. We added a statement about VAP diagnosis in the legend of the figure and in the results section.

---

## [Decision Letter · Decision Letter 1]

7 Jul 2020

Lung ultrasound score to monitor COVID-19 pneumonia progression in patients with ARDS

PONE-D-20-14729R1

Dear Dr. Dargent,

We’re pleased to inform you that your manuscript has been judged scientifically suitable for publication and will be formally accepted for publication once it meets all outstanding technical requirements.

Kind regards,

Muhammad Adrish

Academic Editor

PLOS ONE

Additional Editor Comments (optional):

Reviewers' comments:

Reviewer's Responses to Questions

**Comments to the Author**

1. If the authors have adequately addressed your comments raised in a previous round of review and you feel that this manuscript is now acceptable for publication, you may indicate that here to bypass the “Comments to the Author” section, enter your conflict of interest statement in the “Confidential to Editor” section, and submit your "Accept" recommendation.

Reviewer #1: All comments have been addressed

Reviewer #2: All comments have been addressed

2. Is the manuscript technically sound, and do the data support the conclusions?

Reviewer #1: Yes

Reviewer #2: Yes

3. Has the statistical analysis been performed appropriately and rigorously? 

Reviewer #1: N/A

Reviewer #2: N/A

4. Have the authors made all data underlying the findings in their manuscript fully available?

Reviewer #1: Yes

Reviewer #2: Yes

5. Is the manuscript presented in an intelligible fashion and written in standard English?

Reviewer #1: Yes

Reviewer #2: Yes

6. Review Comments to the Author

Reviewer #1: This is an interesting report about a valuable bedside examination during a time where transport of patients in the hospital has to be limited.

Comments of reviewers have been addressed as feasible with reasonable efforts.

Reviewer #2: All my concerns about the manuscript have been addressed in the revision. The manuscript can be considered for publication. I have no further concerns

7. PLOS authors have the option to publish the peer review history of their article (what does this mean?). If published, this will include your full peer review and any attached files.

Reviewer #1: No

Reviewer #2: No

---

## [Editor Report · Acceptance letter]

13 Jul 2020

PONE-D-20-14729R1 

Lung ultrasound score to monitor COVID-19 pneumonia progression in patients with ARDS 

Dear Dr. Dargent:

I'm pleased to inform you that your manuscript has been deemed suitable for publication in PLOS ONE. Congratulations! Your manuscript is now with our production department. 

Kind regards, 

on behalf of

Dr. Muhammad Adrish 

Academic Editor

PLOS ONE